# Atmospheric Room Temperature Plasma as a Green Pretreatment Strategy for Enhanced Phytochemical Extraction from *Moringa oleifera* Leaves

**DOI:** 10.3390/foods14183233

**Published:** 2025-09-17

**Authors:** Martha Mantiniotou, Vassilis Athanasiadis, Dimitrios Kalompatsios, Eleni Bozinou, George Ntourtoglou, Vassilis G. Dourtoglou, Stavros I. Lalas

**Affiliations:** 1Department of Food Science and Nutrition, University of Thessaly, Terma N. Temponera Street, 43100 Karditsa, Greece; mmantiniotou@uth.gr (M.M.); vaathanasiadis@uth.gr (V.A.); dkalompatsios@uth.gr (D.K.); empozinou@uth.gr (E.B.); 2Department of Wine, Vine and Beverage Sciences, University of West Attica, Agiou Spyridonos Street, Egaleo, 12210 Athens, Greece; gntourtoglou@uniwa.gr (G.N.); vdourt@uniwa.gr (V.G.D.)

**Keywords:** Atmospheric Room Temperature Plasma, ultrasound-assisted extraction (UAE), response surface methodology, bioactive compounds, antioxidant activity, sustainable processing

## Abstract

Over the past few years, naturally sourced bioactive molecules have drawn increased attention for their antioxidant capacity and wide-ranging health effects. At the same time, interest in eco-friendly extraction approaches has risen sharply. Atmospheric Room Temperature Plasma (ARTP), a novel non-thermal pretreatment method, has emerged as a promising green technology due to its minimal environmental impact, cost-effectiveness, and superior extraction efficiency compared to conventional methods. In this study, ARTP pretreatment—optimized across variables such as treatment distance, substrate thickness, power, nitrogen flow, and duration—was combined with ultrasonic-assisted extraction to enhance the recovery of bioactive compounds from *Moringa oleifera* leaves. Both techniques were optimized using Response Surface Methodology (RSM). Under optimal conditions, the extract yielded a total polyphenol content of approximately 40 mg gallic acid equivalents per gram of dry weight. Antioxidant activity, assessed via ferric-reducing antioxidant power (FRAP) and DPPH radical scavenging assays, reached ~280 and ~113 μmol ascorbic acid equivalents per gram dry weight, respectively, and the ascorbic acid content was ~5.3 mg/g. These findings highlight the potential of ARTP as an effective and sustainable pretreatment method for producing high-value phytochemical extracts, with promising applications in the food, pharmaceutical, and cosmetic industries.

## 1. Introduction

The growing demand for natural bioactive compounds in the food, pharmaceutical, and cosmetic industries has intensified the search for extraction methods that are both efficient and environmentally responsible. Conventional techniques often involve high temperatures, organic solvents, and energy-intensive processes that can degrade sensitive compounds and generate harmful waste. In response, non-thermal, green extraction technologies have attracted attention for their ability to preserve bioactivity while minimizing environmental impact. This shift is reinforced by consumer demand for “clean label” products and regulatory pressure to reduce solvent residues and carbon footprints in manufacturing.

In the realm of natural product extraction, pretreatment methods play a pivotal role in maximizing yield and preserving the integrity of bioactive compounds. Atmospheric Room Temperature Plasma (ARTP) has recently gained attention as an innovative pretreatment technology that addresses many limitations of conventional methods. By exposing plant materials to reactive plasma species at room temperature and atmospheric pressure, ARTP disrupts cell walls and alters surface properties, thereby facilitating improved solvent access and mass transfer during extraction [1,2]. It has been successfully used to improve the productivity of biomass and high-value compounds in various food and bioprocessing contexts [3].

The unique advantages of ARTP include low energy consumption, minimal environmental footprint, and versatility across applications ranging from sterilization and wound healing to food processing and biotechnology [4]. In extraction processes, ARTP’s ability to non-thermally disrupt plant cell walls enhances solvent penetration and facilitates the release of intracellular compounds. Additionally, ARTP’s use of nitrogen as a working gas helps minimize oxidative degradation of sensitive bioactives compared to oxygen-rich plasmas. The reactive species generated during ARTP treatment—such as atomic oxygen, ozone, and excited nitrogen molecules—interact with the plant matrix to weaken cellular structures without inducing thermal degradation. This non-thermal mechanism is particularly advantageous for heat-sensitive phytochemicals, as it preserves their biological activity and antioxidant potential [5,6,7]. This aligns with trends in hybrid green extraction technologies, integrating multiple non-thermal methods [8].

In more detail, ARTP operates at ambient temperature and atmospheric pressure, producing a partially ionized gas composed of high energy electrons, ions, ultraviolet photons, and a complex mixture of reactive oxygen species (ROS)—such as ozone (O_3_), atomic oxygen (O), singlet oxygen (^1^O_2_), and superoxide anion (O_2_^−^)—and reactive nitrogen species (RNS), including atomic nitrogen (N), excited nitrogen molecules (N_2_*), and nitric oxide radicals (NO•) [9,10,11,12,13]. In contrast to thermal plasmas, which operate at several thousand kelvins, ARTP can be applied directly to heat-sensitive substrates without thermal degradation. Nitrogen is frequently used as the working gas to bias the reactive chemistry toward RNS while limiting oxidative load, thus safeguarding oxidation-prone phytochemicals [1,14,15,16].

The interaction of these reactive species with plant tissues initiates physical and chemical effects—including etching of epicuticular waxes, depolymerization of cell wall polysaccharides, micropitting of epidermal layers, and increased hydrophilicity of surfaces—all of which facilitate solvent ingress and the release of intracellular metabolites. In food systems, similar mechanisms have been harnessed for microbial inactivation, enzyme inactivation, and surface functionalization [6,7,10]. These multimodal effects are particularly valuable when the target compounds are sequestered within rigid or lignified tissues. Moreover, ARTP’s ability to selectively modify the surface and internal structure of plant tissues enhances the efficiency of subsequent extraction techniques. When combined with ultrasound-assisted extraction (UAE), which uses acoustic cavitation to further disrupt cells and improve solvent penetration [17], ARTP pretreatment creates a synergistic effect that boosts overall extraction yield and quality.

The integration of ARTP into extraction workflows represents a shift towards greener, more sustainable processing methods. It reduces reliance on harsh solvents and energy-intensive steps, aligning with increasing industrial and consumer demand for environmentally responsible production of natural ingredients. By integrating ARTP as a pretreatment step, existing extraction technologies can be augmented without requiring wholesale changes, offering a practical and sustainable solution to improve extraction yields and phytochemical quality. By first modifying cell wall architecture via plasma and then exploiting cavitation-induced shear, the two methods (i.e., ARTP and UAE) can together achieve extraction yields and bioactive recoveries that exceed either technique alone [1,2,5,6,7,13]. Such dual non-thermal strategies have already shown promise for recovering phenolics, carotenoids, and essential oils from various agro-industrial residues and botanical matrices.

*Moringa oleifera* L., of the Moringaceae family, is widely recognized as a “miracle tree” for its nutritional richness, multifunctional utility, and diverse pharmacological attributes [18]. It is used in medicine, cosmetics, lactation enhancement, and culinary contexts, and also serves in water purification, as animal feed, in biostimulants and biopesticides, and as biomass for biodiesel production [19,20,21]. Recent analyses confirm high antioxidant potential across leaves, seeds, flowers, and other tissues [22], but the *M. oleifera* leaves (MOL) are especially valued for their dense phytochemical profile, which includes rutin, kaempferol, quercetin, ferulic acid, myricetin, isorhamnetin, and derivatives [23,24]. These compounds underpin the plant’s antioxidant, antimicrobial, anti-inflammatory, anticancer, and antistress properties [24,25]. However, many of these phytochemicals are sensitive to heat, oxidation, or prolonged processing, which can result in marked activity losses if extracted using conventional thermal or solvent-intensive methods. In the present study, MOL were selected as a model system to evaluate the practical benefits of ARTP pretreatment combined with UAE.

The effectiveness of ARTP pretreatment depends heavily on operational parameters such as plasma power, treatment distance, gas flow rate, substrate thickness, and exposure duration. Each of these factors influences the generation and delivery of reactive species, as well as the degree of interaction with the plant material. Optimizing these variables is crucial for maximizing extraction efficiency while avoiding overexposure, which could lead to excessive degradation or unfavorable chemical modifications.

Given these constraints, integrating a nitrogen-based ARTP pretreatment with optimized UAE offers a promising route to maximizing recovery of MOL phytochemicals while adhering to green processing principles. In this work, Response Surface Methodology (RSM) is used to systematically optimize five ARTP operational parameters—treatment distance, material thickness, plasma power, nitrogen flow rate, and exposure time—in concert with UAE variables. Our objectives were to (i) maximize total polyphenol content (TPC) and antioxidant indices (FRAP, DPPH, ascorbic acid content), (ii) monitor changes in the individual polyphenolic profile by HPLC-DAD, (iii) evaluate colorimetric shifts (CIELAB parameters) that could signal pigment or matrix changes, and (iv) frame the processing outcomes in terms of mechanistic action, operational efficiency, and potential for scale up. Through this approach, optimal ARTP conditions were identified that balance effective cell disruption with preservation of bioactive compounds, demonstrating the technology’s potential for scalable and sustainable extraction processes, and aligning with reports that confirm the feasibility of ultrasound, cold plasma, and innovative solvent systems at pilot and commercial scales [26].

This is, to our knowledge, the first study to report statistically optimized ARTP + UAE for MOL, quantified against UAE-only controls and discussed in the context of both phytochemical enrichment and process sustainability. By detailing both performance metrics and plausible physicochemical mechanisms, we aim to inform the broader adoption of cold plasma pretreatments in plant extraction workflows.

## 2. Materials and Methods

### 2.1. Chemicals and Reagents

L-Ascorbic acid (product no. A7506), trichloroacetic acid (product no. T6399), hydrochloric acid (product no. 258148), 2,2-diphenyl–1-picrylhydrazyl (DPPH^•^) (product no. D9132), 2,4,6-tris(2-pyridyl)-s-triazine (TPTZ)(product no. T1253), aluminum chloride (product no. 237051), iron (III) chloride (product no. 940534), and all chemical standards for the HPLC identification of individual polyphenols were purchased from Sigma-Aldrich (Darmstadt, Germany). Acetonitrile (product no. 608–001–00–3) was purchased from Labkem (Barcelona, Spain). Primary alcohols, including methanol (product no. 131091) and ethanol (product no. 191086), and the Folin–Ciocalteu reagent (product no. 251567) were obtained from Panreac Co. (Barcelona, Spain). Anhydrous sodium carbonate (product no. 28000–31000) and formic acid (98%) (product no. 19920–11000) were obtained from Penta (Prague, Czech Republic). Deionized water generated from a deionizing column was used for spectrophotometric experiments, whereas ultrapure water (resistivity 0.055 µS/cm) was used for chromatographic determinations.

### 2.2. Plant Material

The central Greek area of Krya Vrisi (Karditsa region, Greece) was the site of the *M. oleifera* seedling cultivation. The coordinates were 39°19′6.97″ N and 21°52′39.16″ E, at an elevation of 131 m. More details about this region can be found in our previous works [18,27]. After the MOLs were harvested, they were transported to the laboratory within 30 min. Upon arrival, the leaves were thoroughly rinsed with tap water and subjected to an overnight freeze-drying process at −54 °C using a BK-FD10P lyophilizer from Biobase (Jinan, China), followed by deep-freezing at −40 °C. Finally, they were ground to a fine powder using an electric ball mill to a particle size of 400–600 μm and stored at −40 °C until further analysis.

### 2.3. Instruments and Software

An EWJ 600–2M precision scale from Kern (Frankfurt, Germany) was used to weigh the freeze-dried and ground MOL.

The ARTP process was conducted using the ARTP apparatus, produced via Piezoelectric Direct Discharge (PDD^®^) technology using the Piezobrush^®^ PZ3-i device from Relyon Plasma GmbH (Regensburg, Germany). The apparatus was connected to a handmade adjustable power supply to regulate/control the energy produced by the device. After the ARTP process, the extracts, including solvent and MOL, were placed in polypropylene centrifugal tubes. To separate the liquid supernatant from the solid byproduct, a NEYA 16R centrifuge from Remi Elektrotechnik Ltd. (Palghar, India) was used to spin the samples at 4500 rpm for 5 min.

Ultrasonication (US) was performed with an Elmasonic P70H bath from Elma Schmidbauer GmbH (Singen, Germany).

The spectrophotometric analyses were conducted using a UV–1900i PharmaSpec double-beam spectrophotometer manufactured by Shimadzu (Kyoto, Japan). An HPLC system by Shimadzu Europa GmbH (Duisburg, Germany) was used for the chromatographic analyses. To identify individual polyphenols, a CBM–20A HPLC model was linked to an SPD-M20A diode array detector (DAD). The identified compounds were separated using a Luna C18(2) column (100 Å, 5 μm, 4.6 mm × 250 mm) supplied by Phenomenex Inc. (Torrance, CA, USA). The Lovibond CAM-System 500 colorimeter, manufactured by The Tintometer Ltd. (Amesbury, UK), was employed to ascertain the CIELAB parameters (*L**, *a**, and *b**) from the MOL extracts.

Google Earth was employed for the determination of geographical coordinates from Google Inc. (Cambridge, MA, USA). Data analysis and statistical processing were carried out using JMP^®^ Pro 16 software (SAS, Cary, NC, USA). A schematic diagram illustrating the combined use and sequential arrangement of all instruments is presented in Figure 1 to facilitate understanding of the experimental workflow.

### 2.4. Ultrasound-Assisted Extraction of Phytochemicals: Design and Optimization Framework

To optimize the phytochemical extraction process, the TPC was selected as the primary response metric, providing a quantitative measure of both extract yield and antioxidant capacity. This parameter served as a robust basis for assessing the influence of UAE variables.

The study applied RSM using a Custom Design to identify optimal conditions for TPC extraction from dried MOL. Ultrasonication was conducted at 37 kHz and maintained at a constant temperature of 30 °C.

Four independent variables were evaluated: The liquid-to-solid ratio (*R*, mL/g) as *X*_1_, the ethanol concentration in water (*C*, % *v*/*v*) as *X*_2_, the ultrasound energy (*E*, %) as *X*_3_, and the extraction time (*t*, min) as *X*_4_. Each variable was tested at three levels—low (−1), medium (0), and high (+1)—as detailed in Table 1. A total of 15 experimental runs were conducted, with each run performed in triplicate, and the mean TPC values used for analysis.

A second-order polynomial Equation (1) was developed using the method of least squares to model the interactions among the variables:(1)$$Y_{k}\;=\;\beta_{0}+\sum_{i=1}^{2}\beta_{i}X_{i}+\sum_{i=1}^{2}\beta_{\mathrm{ii}}X_{i}^{2}+\sum_{i=1}^{2}\sum_{j=i+1}^{3}\beta_{\mathrm{ij}}X_{i}X_{j}$$
where *Y_k_* represents the predicted TPC response; *X_i_* and *X_j_* are the coded independent variables; *β*_0_ is the intercept; and *βᵢ*, *βᵢᵢ*, and *β_ij_* denote the linear, quadratic, and interaction coefficients, respectively.

RSM facilitated the identification of conditions that maximize TPC yield while elucidating the individual and interactive effects of each extraction factor. Contour plots were constructed to visualize the interactions among variables, thereby enhancing the identification of optimal extraction conditions.

### 2.5. Determination of Phytochemicals Compounds

#### 2.5.1. Polyphenolic Compounds Determination and Quantification

The assessment of TPC relied on the previously established methods [28]. The reduction in a mixture containing phosphotungstic acid and phosphomolybdic acid from polyphenolic compounds resulted in a mixture of blue oxides where the spectrophotometric measurement at 740 nm was recorded. The results were expressed in terms of milligrams of gallic acid equivalents (GAE) per gram of dry weight (dw), as indicated in Equation (2). The dry weight of the sample (*w*, in g), the exact volume of the extraction solvent (*V*, in L), and the total polyphenol content (*C*_TP_, mg/L) are all computed. The analysis was preceded by the execution of a calibration curve (10–100 mg GAE/L in methanol).(2)$$\mathrm{TPC}\;(\mathrm{mg}\;\mathrm{GAE}/g\;\mathrm{dw})=\frac{C_{\mathrm{TP}}\;\times \;V}{w}$$

High-Performance Liquid Chromatography (HPLC) analyses were performed using a Shimadzu Europa GmbH (Duisburg, Germany) HPLC system (CBM–20A model) equipped with an SPD-M20A diode array detector (DAD). Separation of individual polyphenols was achieved on a Luna C18(2) column (100 Å, 5 µm, 4.6 mm × 250 mm; Phenomenex Inc., Torrance, CA, USA) maintained at 40 °C. The mobile phase consisted of solvent A (0.5% *v*/*v* formic acid in water) and solvent B (0.5% *v*/*v* formic acid in acetonitrile), delivered at a flow rate of 1.0 mL/min. The injection volume was 20 µL. The gradient program (total run time: 85 min) involved a linear increase from 0% to 40% B, followed by 50% B for 10 min, then 70% B for 10 min, and a final 10 min hold. Samples were diluted according to TPC data to fit within the calibration range. Polyphenols were identified by comparing retention times and UV–Vis spectra with pure standards and quantified using calibration curves (0–50 mg/L, R^2^ > 0.99). Concentrations were expressed as mg/g dry weight, and retention time, λ_max_, limit of detection (LOD), and limit of quantification (LOQ) for each compound are provided in Table A1.

#### 2.5.2. Total Flavonoid Content (TFC)

The total flavonoid content (TFC) was determined through a complexometric reaction between aluminum chloride and flavonoids from the sample, using a previously published method [27]. Volume of sample was mixed with 35% *v*/*v* aqueous ethanol and 5% *w*/*v* aluminum chloride in a ratio of 10:86:4. The absorbance of the formed yellowish mixture was recorded at 415 nm after 30 min of storage at room temperature. TFC was evaluated through the determination of total flavonoid concentration (*C*_TFn_) using a quercetin calibration curve (30–300 mg/L in methanol), as indicated in Equation (3). The results were expressed as mg of quercetin equivalents (QE) per g of dw.(3)$$\mathrm{TFC}\;(\mathrm{mg}\;\mathrm{QE}/g\;\mathrm{dw})=\frac{C_{\mathrm{TFn}}\;\times \;V}{w}$$

#### 2.5.3. L-Ascorbic Acid Content (AAC)

L-Ascorbic acid content (AAC) was determined in accordance with a previous methodology [29]. This method relies on the redox reaction between 10-fold diluted Folin–Ciocalteu reagent and the specific antioxidant compound, acidified with 10% *w*/*v* aqueous trichloroacetic acid, where the absorbance at 760 nm of a greenish mixture is recorded. Results are reported in milligrams of ascorbic acid per gram of dry weight, calculated using a calibration curve prepared over 50–500 mg/L AA.

### 2.6. Antioxidant Capacity Evaluation

#### 2.6.1. Ferric Reducing Antioxidant Power (FRAP) Assay

The antioxidant capacity of MO leaf extracts was evaluated using an ion-reducing test, as established by a previous study [28]. Following Fe(III) reduction and its subsequent complexation with TPTZ, a stable bluish Fe^2+^–TPTZ complex formed, and its absorbance was measured at 620 nm. Prior to analysis, a calibration curve containing ascorbic acid dissolved in 0.05 M HCl was utilized (ranging from 50 to 500 μM). Equation (4) was used to evaluate the ferric ions reducing power (*P*_R_), which was displayed as μmol of ascorbic acid equivalents (AAE) per g of dry weight (dw).(4)$$P_{R}\;(\mu \mathrm{mol}\;\mathrm{AAE}/g\;\mathrm{dw})=\frac{C_{AA}\;\times \;V}{w}$$

#### 2.6.2. DPPH Antiradical Activity (A_AR_)

MOL extracts were further tested for their antioxidant properties using a radical scavenging assay. Each extract was combined with methanolic DPPH^•^ (100 μM) at a 39:1 (DPPH–sample extract) ratio, following the method outlined earlier by Kalompatsios et al. [29]. The spectrophotometric evaluation of DPPH^•^ decolorization was performed at 515 nm. Equation (5) was used to calculate the % inhibition from the initial absorbance (*A*_515(i)_) and final absorbance (*A*_515(f)_) after 30 min of storage in the dark. Using a robust antioxidant with scavenging capability (i.e., AA), it was possible to calculate the antiradical activity (*A*_AR_) of each extract. The variables evaluated were the volume of solvent (*V*, in L), the concentration of AA (*C*_AA_, in μmol), and the quantity of MOL (*w*, in g). Antiradical activity (*A*_AR_) of MOL extracts was expressed in μmol of AAE/g dw, as per Equation (6).(5)$$\mathrm{Inhibition}\;\left(\%\right)=\frac{A_{515\left(i\right)\;}-A_{515\left(f\right)}}{A_{515\left(i\right)}}\;\times \;100$$(6)$$A_{\mathrm{AR}}\;\left(\mu \mathrm{mol}\;\mathrm{AAE}/g\;\mathrm{dw}\right)=\frac{C_{\mathrm{AA}}\;\times \;V}{w}$$

### 2.7. Color Analysis

Color determination of MOL extracts involved recording CIELAB coordinates (*L**, *a**, *b**) following an established protocol [27]. The *L** coordinate denotes lightness, spanning from 0 (black) to 100 (white), while the *a** axis indicates the red–green dimension, with positive values toward red and negative toward green. Colors can also be evaluated by their *b** values, which indicate how much they lean toward yellow (positive values) or blue (negative values), respectively.

### 2.8. Plasma Treatment Experimental Design

To optimize phytochemical recovery from MOL, RSM with a Central Composite Design (CCD) was applied. This statistical approach targeted multiple response variables: TPC, FRAP, DPPH^•^ antiradical activity, and AAC.

The pretreatment procedure employed ARTP and PDD processes, which produce plasma by emitting high electric fields from a piezo-ceramic transformer, converting a low input voltage (24 V DC) into high-voltage discharge within a process gas. Nitrogen was used as the working medium to activate reactive nitrogen species, including charged particles, radicals, and UV photons.

Its compact, modular design enables seamless integration into standard laboratory workflows. It operates with a maximum power consumption of 18 W under ambient conditions (~24 °C, 40–45% relative humidity). Plasma treatments were performed using the Standard Module, optimized for non-conductive substrates including plastics and ceramics. The use of the Standard Module was appropriate for treating MOL powder, as both the plant material and polystyrene culture dishes are non-conductive substrates, aligning with the module’s operational design parameters. All procedures were conducted within a ventilated fume hood to ensure operator safety.

For each run, MOL powder was placed into polystyrene culture dishes (35 mm × 10 mm) and leveled with a spatula to achieve a uniform surface thickness. This setup ensured consistent exposure during plasma treatment and reproducibility across experimental runs.

Five independent variables were evaluated to define the pretreatment efficacy: The distance between material surface and plasma source (*D*, mm) as *X*_1_, the material thickness (*T*, mm) as *X*_2_, the plasma power level (*P*, %) as *X*_3_, the nitrogen flow rate (*N*, L/min) as *X*_4_, and the exposure time (*t*, min) as *X*_5_. Each variable was tested at three levels—low (−1), medium (0), and high (+1)—as presented in Table 2. A total of 28 experimental runs, including two central points, were performed in triplicate. The mean values of each response were recorded for model fitting and analysis.

Following ARTP pretreatment, each powder sample underwent UAE under previously optimized conditions to recover target phytochemicals.

To enhance predictive accuracy, stepwise regression was applied to eliminate non-significant terms and minimize model variance. As in Section 2.4, a second-order polynomial equation was employed to model the interactions among the independent variables. The equation structure remains unchanged, though the variables and response metrics differ, as outlined in Table 2. This model enabled the quantitative evaluation of each ARTP variable’s influence and supported optimization for enhanced phytochemical extraction yields.

### 2.9. Statistical Analysis

Each batch of MOL extracts was analyzed in triplicate (*n* = 3), and the full extraction procedure was performed in duplicate (two independent batches). Data normality was examined using the Kolmogorov–Smirnov test. One-way analysis of variance (ANOVA) was used to find the differences that were statistically significant. The results are shown by means and standard deviations. Partial Least Squares (PLS), Multivariate Correlation Analysis (MCA), Principal Component Analysis (PCA), and Pareto Plot Analysis were used to assess the degree of correlation between the parameters that were being studied.

## 3. Results and Discussion

### 3.1. Optimization of the Extraction Process of Phytochemicals

The first step to finding the optimal extraction conditions was the optimization of the UAE. Four important parameters such as liquid-to-solid ratio (*X*_1_), ethanol concentration (*X*_2_), ultrasound energy (*X*_3_) and extraction duration (*X*_4_) were studied. TPC was used to evaluate extraction efficiency. Our RSM-derived conditions align with recent optimization studies of UAE for phenolic and flavonoid recovery [17]. In Table 3, the values predicted by the model as well as the actual TPC values obtained experimentally are displayed. It is clear that the model has an excellent fit to the experimental data, as the experimental values are not statistically significantly different from the predicted ones. Table 4 reports the RSM ANOVA, supporting model adequacy (R^2^ = 0.9999). Equation (7) clarifies regression models pertinent to the extraction process, forecasting the principal response variable, TPC. The intricate relationships among experimental variables are emphasized through the inclusion of linear, quadratic, and interaction terms in each equation. The models consisted solely of significant terms, while non-significant terms were retained only if they improved overall model fit. The regression models emphasize the influence of solvent composition, ultrasonic power, liquid-to-solid ratio, and extraction duration on extraction efficiency. The linear and quadratic terms signify non-linear relationships among variables, indicating the presence of optimal conditions for maximum antioxidant output. The extraction conditions, particularly the liquid-to-solid ratio (*X*_1_) and the ethanol concentration (*X*_2_), substantially influence the TPC equation.*TPC* = −24.95 − 0.19*X*_1_ + 0.49*X*_2_ + 0.46*X*_3_ + 4.84*X*_4_ − 0.0004*X*_1_^2^ − 0.004*X*_2_^2^ − 0.098*X*_4_^2^ + 0.001*X*_1_*X*_2_ + 0.002*X*_1_*X*_3_ + 0.018*X*_1_*X*_4_ − 0.002*X*_2_*X*_3_ − 0.006*X*_2_*X*_4_ − 0.036*X*_3_*X*_4_(7)

Contour plots are a powerful graphical technique that visualizes three-dimensional interactions on a two-dimensional plane. By rendering complex interactions as topographical visuals, these plots offer intuitive insights into variable relationships. This analysis process enables quick recognition of interaction effects between variables. In Figure 2A, the covariation of *X*_1_ and *X*_2_ is depicted, and it is obvious that higher liquid-to-solid ratios necessitate lower ethanol concentrations. Similarly, Figure 2B,C depict the covariations between *X*_1_ and *X*_3_ and *X*_1_ and *X*_4,_ respectively, and it is concluded that higher ratios have a better impact when higher US energy and extraction durations are applied. In Figure 2D, it is obvious that average ethanol concentrations are more efficient at higher US energy is applied, while Figure 2E,F indicate that there are no significant interactions between *X*_2_ and *X*_4_ and between *X*_3_ and *X*_4_.

In Figure 3, the optimal extraction conditions for UAE, along with the predicted optimal TPC and the desirability, according to the RSM desirability function. The optimal liquid-to-solid ratio was established as 70 mL/g, the optimal ethanol concentration 35% *v*/*v*, the optimal US energy 60% and the optimal duration 15 min.

The predicted TPC value of 37 ± 3 was experimentally validated under optimized extraction conditions, resulting in a measured value of 34.7 ± 0.6, demonstrating strong agreement and validating the predictive model’s robustness. UAE optimization was of paramount importance, as at all experimental sites where ARTP will be applied, the bioactive compounds will be recovered through optimal UAE conditions. In addition, the optimal UAE will be used as a control in order to evaluate whether the application of ARTP has positive or negative effects on the extraction of bioactive compounds from MOL.

### 3.2. Optimization of the Plasma Treatment

The next step was to perform the optimization of the ARTP conditions. The main parameters studied were the Distance between the material surface and the plasma source (*X*_1_), the material thickness (*X*_2_), the plasma power level (*X*_3_), the nitrogen flow rate (*X*_4_), and the exposure time (*X*_5_). For ARTP optimization, the responses used to evaluate the results were TPC, FRAP, DPPH, and AAC. In Table 5, the design points of the experimental design and their actual responses are illustrated, while Table 6 summarizes the ANOVA applied to the RSM quadratic polynomial model.

### 3.3. Model Optimization

Regression models pertaining to the extraction process are shown in Equations (8)–(11). These models predict important response variables such as TPC, FRAP, DPPH, and AAC. The models contain only significant terms, and they indicate how each variable affects the extraction efficiency. The non-linear relationship between variables, as indicated by the linear and quadratic terms, reveals conditions that maximize antioxidant yield. The presence of interaction terms indicates that the cumulative effects of various variables affect antioxidant potential, highlighting the necessity for meticulous parameter optimization. Extended extraction durations facilitate the dissolution of additional bioactive compounds, including antioxidants, into the solvent. The presence of quadratic terms (such as *X*_1_^2^, *X*_2_^2^, etc.) and interactions (*X*_1_ × *X*_2_, *X*_1_ × *X*_3_, *X*_4_ × *X*_5_, etc.) implies that extraction time possesses an optimal range—insufficient duration may restrict compound release, whereas prolonged periods could result in degradation or diminished efficiency. The interaction terms suggest that extraction time operates in conjunction with other factors. The models contained only significant terms. Equations:*TPC* = 27.53 + 0.52*X*_1_ − 0.73*X*_2_ + 0.04*X*_3_ + 0.89*X*_4_ − 0.81*X*_5_ − 0.01*X*_1_^2^ + 0.19*X*_2_^2^ − 0.03*X*_4_^2^ + 0.04*X*_5_^2^ − 0.030*X*_1_*X*_2_ − 0.002*X*_1_*X*_3_ − 0.009*X*_4_*X*_5_(8)*FRAP* = 172.95 + 1.76*X*_1_ + 0.24*X*_2_ + 1.29*X*_3_ + 3.29*X*_4_ − 5.28*X*_5_ + 2.44*X*_2_^2^ − 0.01*X*_3_^2^ + 0.26*X*_5_^2^ − 0.270*X*_1_*X*_2_ − 0.029*X*_1_*X*_4_ + 0.066*X*_2_*X*_3_ − 0.734*X*_2_*X*_4_ − 0.217*X*_2_*X*_5_ + 0.032*X*_4_*X*_5_(9)*DPPH* = −6.83 + 2.72*X*_1_ + 1.84*X*_2_ − 0.93*X*_3_ + 14.80*X*_4_ − 0.51*X*_5_ − 0.05*X*_1_^2^ + 1.77*X*_2_^2^ + 0.005*X*_3_^2^ − 0.44*X*_4_^2^ + 0.05*X*_5_^2^ − 0.081*X*_1_*X*_2_ − 0.006*X*_1_*X*_3_ + 0.021*X*_1_*X*_5_ + 0.042*X*_2_*X*_3_ − 0.808*X*_2_*X*_4_ − 0.259*X*_2_*X*_5_ + 0.015*X*_3_*X*_4_ + 0.010*X*_3_*X*_5_ − 0.051*X*_4_*X*_5_(10)*AAC* = 2.44 + 0.13*X*_1_ − 0.37*X*_2_ − 0.02*X*_3_ + 0.33*X*_4_ − 0.02*X*_5_ − 0.003*X*_1_^2^ + 0.073*X*_2_^2^ + 0.0001*X*_3_^2^ − 0.009*X*_4_^2^ + 0.003*X*_5_^2^ + 0.0011*X*_1_*X*_2_ + 0.0002*X*_1_*X*_3_ − 0.0009*X*_1_*X*_4_ − 0.0061*X*_2_*X*_4_ − 0.0051*X*_2_*X*_5_ − 0.0003*X*_3_*X*_4_ − 0.0004*X*_3_*X*_5_(11)

Figure 4, Figure 5, Figure 6 and Figure 7 display contour plots for TPC, FRAP, DPPH, and AAC. Figure 4A indicates that the higher the distance between the material surface and the plasma source, the lower the exposure time should be to be more efficient for higher TPC yield. A similar pattern is observed in Figure 4B,C, regarding the distance between material surface and plasma source and plasma power level and nitrogen flow rate and exposure time, respectively. In Figure 5 it is indicated that FRAP gives higher results when combinations of the extreme values of the variables under consideration are applied. On Figure 6 it is shown that the combinations of all the values of the variables do not significantly affect the performance of DPPH. A similar conclusion can be drawn for the AAC in Figure 7.

Table 7 provides the model’s proposed optimal covariates for each response separately, along with the model desirability and the optimal values obtained by stepwise regression. All models show excellent desirability (>0.9), indicating how correctly the selected statistical model has been applied to the data. Regarding parameter *X*_1_, TPC and FRAP require the highest value (e.g., 30 mm), while DPPH and AAC require an intermediate value (24 and 20 mm, respectively). For parameter *X*_2_, all responses are in agreement that the optimal condition requires 1 mm. When it comes to plasma energy (parameter *X*_3_), it appears that each response demands different energy, as only TPC and AAC concur with the lowest possible existing energy, FRAP slightly higher energy, while DPPH demands the highest plasma energy to drive the maximum performance. Regarding the nitrogen flow (*X*_4_), all responses settle at the same value of 17 L/min, with only a slight variation in FRAP requiring 19 L/min. Finally, considerable differences are observed in the process duration (*X*_5_), as TPC and FRAP lead to maximum yield in only 1 min, while DPPH and AAC lead to the maximum yield in the highest possible duration of 19 min.

### 3.4. Impact of Extraction Parameters: Pareto Plot Analysis

In the Pareto plot (Figure 8), an orthogonal estimate refers to a statistical technique used to evaluate the impact of various factors while reducing interdependence among them. This method enhances the clarity of results by producing estimates that are statistically independent, making it easier to pinpoint the most influential variables. Orthogonal estimates are commonly employed in regression and experimental design to increase the precision of parameter calculations, minimize estimation bias, and prevent distortions caused by interactions among different factors. According to Figure 8, TPC is positively affected by the duration of the process, the distance of the surface from the plasma source, and the thickness of the material. Covariations between different parameters (e.g., *X*_1_ × *X*_2_) have a negative effect on TPC. FRAP follows a similar pattern to TPC, with a significant difference in that the combination of parameters *X*_2_ and *X*_4_ has a negative effect on performance. DPPH is also in agreement with FRAP, with a notable difference that the *X*_4_ parameter exerts a significantly negative effect on performance. Finally, AAC is positively influenced mainly by material thickness, while the combination of plasma energy and time has a negative effect on performance.

### 3.5. Principal Component and Multivariate Correlation Analysis

A PCA plot was implemented to enable a more thorough analysis of the data and to extract additional insights from the variables under investigation, as shown in Figure 9. The model accounted for 77.3% of the variance. The purpose of the correlation analyses was to determine the presence of any associations between TPC, FRAP, DPPH, and AAC and the variables under investigation. In this context, the effects of the independent variables on these measures were deemed statistically significant. DPPH and AAC have a positive correlation with *X*_1_ and *X*_4_ parameters, while FRAP and TPC have a moderate positive correlation. This may be due to the fact that the combination of nitrogen flow and the distance between the material surface and the plasma source may lead to the generation of reactive forms of nitrogen that are capable of disrupting MOL cell membranes to a sufficient extent to recover bioactive compounds and ascorbic acid, thus enhancing the extraction efficiency. All four responses display a negative correlation with parameters *X*_2_, *X*_3,_ and *X*_5_.

The MCA offers additional insights into the relationships among variables. This approach primarily enables the assessment of the extent of positive or negative correlation between the variables being studied. The results of MCA are presented in Table 8. According to MCA, all responses have a poor correlation with one another, especially when it comes to TPC and DPPH (~0.2).

### 3.6. Partial Least Squares (PLS) Analysis

A PLS model was employed to determine the significance of the extraction parameters and to locate the optimal ones. In Figure 10, the correlation loading map is presented, which distinctly depicts the extraction circumstances of MOL, as it was formed according to the PLS. In Plot B, the Variable Importance Plot (VIP) is depicted, emphasizing the significance of each predictor variable in the ARTP pretreatment. The red dashed line indicates the 0.8 significance level, highlighting the comparative influence of each variable inside the model. The VIP indicates that the parameters primarily influencing performance are mostly covariations of the examined parameters, such as *X*_2_ × *X*_2_, *X*_5_ × *X*_5_, *X*_4_ × *X*_4_, etc. The covariation of material thickness and nitrogen flow (*X*_2_ × *X*_4_) received the highest score. In PLS, the effects of the parameters on extraction efficiency are distinctly illustrated, facilitating the establishment of optimal conditions. The optimal conditions for ARTP pretreatment on MOL were established as follows: 25 mm distance between plasma source and material surface, 1 mm material thickness, 30% plasma energy, 17 L/min nitrogen flow, and 1 min treatment duration. The model’s fit was satisfactory, as it possesses a desirability of 0.87.

In Table 9, the optimal values of the responses are presented, along with those of the control sample. ARTP process scored an optimal TPC value of 43.29 mg GAE/g dw, which was ~25% higher than the control MOL extract. This value was even higher than the one predicted by the PLS regression. This increase is considerable, and it is also compelling evidence that the process had a positive effect on the recovery of polyphenols from MOL. In our previous study [27], where MOL was subjected to pressurized liquid extraction, the optimal TPC was ~78% lower, thus enhancing the claim that ARTP is suitable for the recovery of bioactive compounds. In another study [30], where MOL was subjected to extraction solvent, the ethanolic extract yielded a TPC of 39.7 mg GAE/g dw after 60 days of extraction. In our study, a TPC 9% higher in just 1 min of treatment. Pollini et al. [31] also utilized the UAE to recover antioxidant compounds from MOL, but they yielded a TPC 3 times lower than ours. Regarding the antioxidant assays, FRAP and DPPH, an increase of 14% and 15.4% was observed from the control, respectively. The optimal ARTP FRAP value was 276.03 μmol AAE/g dw, once again slightly higher than the PLS-predicted value. This value is considered quite satisfactory, considering that lower rates have been obtained in already published papers. Karageorgou et al. [32] established a hydroethanolic extract of MOL, a FRAP value of ~131 μmol AAE/g dw, significantly lower than the one in the current study. Moreover, Karthivashan et al. [33] also determined the antioxidant activity of hydroethanolic MOL extracts through FRAP, and their recovery was significantly lower. Ascorbic acid content presented a 6.3% increase through ARTP. In our previous study [27], through pressurized liquid extraction, a value ~15% lower was determined, thus enhancing the better performance of ARTP at MOL.

From a processing standpoint, the ARTP module employed here has a modest energy demand (18 W) and achieves optimal enhancement of phytochemical recovery within 1 min of treatment. Compared with literature reports of conventional maceration, which can require up to 60 days for similar matrices, this represents a step-change in throughput and a substantial reduction in energy and solvent use (Table 9). Such operational efficiency underscores ARTP’s potential as a rapid, low-resource intervention readily adaptable to continuous-flow or batch industrial settings. It also offers promising scalability and integration potential in existing processing lines [3].

A high correlation coefficient of 0.998 demonstrates strong concordance between experimental results and PLS model predictions and a notable coefficient of determination (R^2^) of 0.996. Furthermore, the minimal *p*-value (<0.0001) confirms that the differences between observed and predicted values are statistically insignificant.

### 3.7. Comparative Evaluation of Polyphenolic Profiles

Figure 11 illustrates a comparative bar chart of (A) total flavonoid content (TFC) and (B) CIELAB color coordinates (*L**, *a**, *b**) in control and optimized samples. The optimal extract demonstrated a substantial increase in TFC (39.4 ± 1.0 mg QE/g dw) versus the control (21.6 ± 0.5 mg QE/g dw), with statistically significant differences (*p* < 0.05) and an 82.6% increase. The *L** values, representing lightness, remained relatively unchanged, indicating minimal variation in brightness. However, *a** shifts from a slightly red hue in the control (0.4) to a greener tint in the optimal sample (−1.2), and *b** values decrease from a vivid yellow (31.2) to a more subdued yellow (22.4), revealing noticeable chromatic differences. These variations suggest possible pigment modifications or compound interactions associated with increased flavonoid content. A possible explanation could lie in changes in the stereochemistry of compounds that contribute to the specific color of MOL (i.e., chlorophyll). A future study further studying the mechanism behind the ARTP procedure could shed more light on that hypothesis.

Comparable pigment-related effects of ARTP have been documented in microalgae. Liu et al. [34] reported that an ARTP-derived *Auxenochlorella pyrenoidosa* mutant (A4–1) exhibited a golden-yellow biomass color, with chlorophyll a reduced 118-fold and chlorophyll b undetectable relative to the wild type. They attributed this to inhibition of chlorophyll biosynthesis, likely via mutations in key enzymes such as light-independent protochlorophyllide oxidoreductase. While our color data are expressed in CIELAB coordinates for ethanolic extracts, rather than direct pigment quantification in biomass, both cases reflect plasma-driven pigment modification. The substantial *a** reduction and *b** decrease in the MOL extracts may similarly arise from partial chlorophyll degradation or altered pigment–matrix interactions induced by ARTP pretreatment.

In Table 10, the individual polyphenolic compounds in the control and optimal extracts determined by HPLC-DAD are presented. Figure 12 depicts a representative chromatograph of these extracts, and in Table A1, the equations of calibration curves of the compounds, along with R^2^, retention times, UV_max_, LOD, and LOQ are provided. Several phenolic acids and flavonoids were determined in both MOL extracts, the control, and the optimal ARTP. In all identified compounds, an increase is observed in the case of ARTP pretreatment, with the most significant being that of quercetin 3-β-*D*-glucoside, where an increase of ~181% was facilitated. In general, all polyphenolic compounds had a significant increase when ARTP was the case. The most abundant polyphenolic compound identified in both control and optimal MOL extracts, neochlorogenic acid, presented a ~51% increase in its recovery. Braham et al. [35] and Rodríguez-Pérez et al. [36] also determined this compound on MOL. Xu et al. [37] also verify the presence of 3-caffeoylquinic acid, or else chlorogenic acid, on MOL. Other researchers [25,38,39] also highlighted the presence of chlorogenic acid, kaempferol–3-glucoside (astragalin), quercetin and derivatives, myricetin, etc., in MOL. According to Leone et al. [40], MOL subjected to water extraction at 80 °C only yielded 0.489 mg chlorogenic acid/g. In another study [41], the presence of isorhamnetin–3-*O*-glucoside was also verified. Pollini et al. [31] found an astragalin content of 0.218 mg/g dw in the UAE MOL extract, which emphasizes the excellent performance of ARTP pretreatment, as it led to almost 6.5 times increased yield of this compound.

The enhanced recovery of TPC and TFC under optimal ARTP pretreatment is likely attributable to the action of reactive nitrogen species (RNS) generated by N_2_ plasma. In mutagenesis applications, ARTP induces genetic changes, whereas in pretreatment workflows such as the present study, the primary effect is physical and chemical modification of cell walls. These species can etch and partially oxidize cell wall polymers, weakening structural barriers and facilitating solvent penetration, while the comparatively low generation of reactive oxygen species (ROS) minimizes oxidative degradation of phenolics. This dual effect is consistent with the observed pigment shifts (*a** toward green, lower *b** values) and the marked increases in polyphenol yield and antioxidant indices. Although outside the scope of the present work, microscopy techniques such as scanning or transmission electron microscopy could provide valuable visual confirmation of the hypothesized cell wall modifications induced by ARTP.

Beyond its technical performance, ARTP pretreatment in our setup also demonstrated operational cost-efficiency. Each batch (1 min treatment) consumed approximately 0.0003 kWh of electricity (18 W power draw) and 0.017 m^3^ of nitrogen gas, resulting in negligible energy costs and a low gas expense at standard laboratory supply rates. No additional consumables, such as enzymes or catalysts, were required. In contrast, conventional enzymatic or thermal pretreatments often involve higher energy demands and/or costly reagents. These factors further support the practicality of ARTP as a pretreatment method for MOL extraction.

While these advantages are notable, one potential trade-off of ARTP pretreatment is the chromatic shift toward greener (*a**) and less yellow (*b**) hues, which may influence consumer perception for applications where color is a key quality cue. Further work should assess whether such changes are acceptable or advantageous in the intended product category.

Addressing such sensory impacts, alongside further mechanistic studies, will be important in guiding industrial adoption. Follow-on studies should explore ARTP’s applicability across diverse botanical sources (e.g., spent MOL, stems) and its integration with other emerging extraction technologies. Such efforts will further establish cold plasma pretreatment as a cornerstone of sustainable and efficient extraction in the food, pharmaceutical, and cosmetic industries, supporting the broader application of the technology and guiding optimal integration into existing production chains.

## 4. Conclusions

In this study, the performance of ARTP as a pretreatment technique for the recovery of bioactive and antioxidant compounds from MOL was investigated. The results were mainly compared with an optimized UAE in order to highlight the performance of the pretreatment technique. The significant increases in performance observed in extraction with ARTP pretreatment arise from the unique physicochemical effects induced by cold plasma. The production of ionized particles was confirmed not to cause degradation of the target compounds. On the contrary, these particles simply helped in the disruption of MOL cell membranes with the sole result of their increased recovery. This process is simple, easy, fast, and environmentally friendly. The maximum performance in MOL required only 1 min of the process, with the plasma energy of 30%, the nitrogen flow at 17 L/min, the distance between plasma source and material surface at 25 mm, and the material thickness at 1 mm. These results are remarkable, as they pave the way for ARTP to become widely used in the food industry. So far, the scientific community has focused on the pharmaceutical uses of the process. However, it is now becoming possible to gain additional applications in various sectors, such as the food and feed industries. Further integration with other eco-efficient processes could enhance yield and sustainability.

## Figures and Tables

**Figure 1 foods-14-03233-f001:**
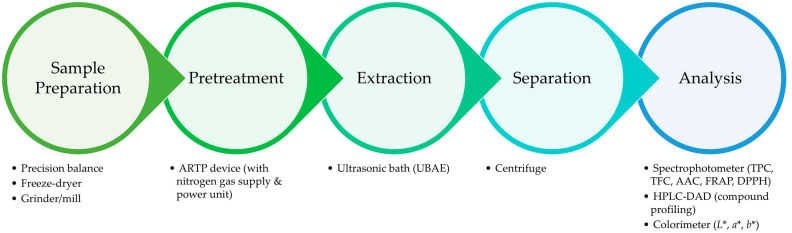
Schematic representation of the instruments used in this study and their sequential integration in the ARTP pretreatment, UAE, and analytical workflow.

**Figure 2 foods-14-03233-f002:**
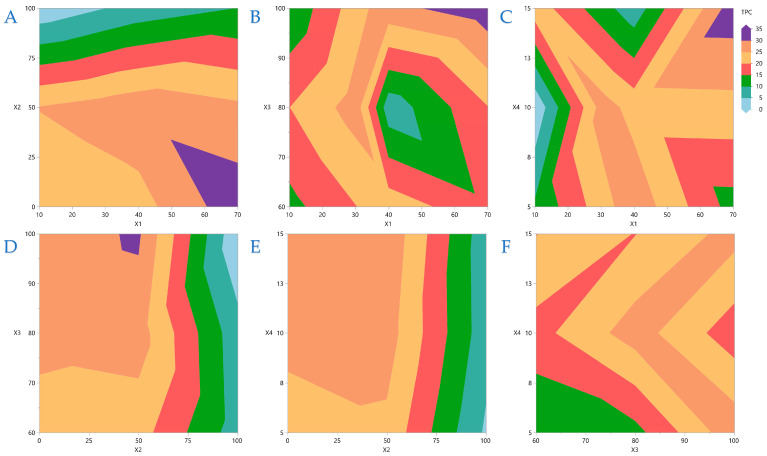
Contour plots for TPC (mg GAE/g dw): plot (**A**) represents the covariation of *X*_1_ (liquid-to-solid ratio) and *X*_2_ (ethanol concentration in water); plot (**B**) represents the covariation of *X*_1_ and *X*_3_ (ultrasound energy); plot (**C**) represents the covariation of *X*_1_ and *X*_4_ (extraction time); plot (**D**) represents the covariation of *X*_2_ and *X*_3_; plot (**E**) represents the covariation of *X*_2_ and *X*_4_; plot (**F**) represents the covariation of *X*_3_ and *X*_4_.

**Figure 3 foods-14-03233-f003:**
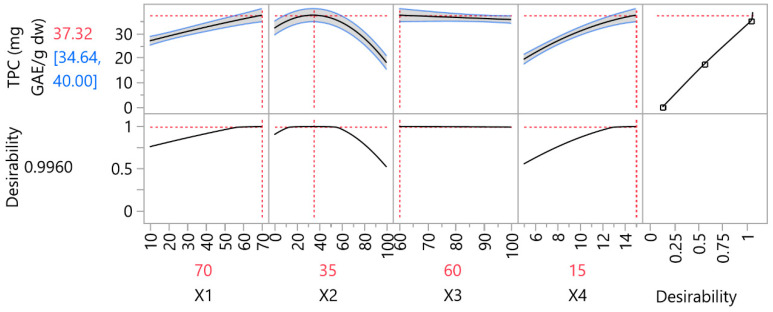
Prediction profiler for the extraction process of phytochemicals. The red number (e.g., 37.32) represents the predicted TPC at the optimal point. The blue interval [34.64, 40.00] shows the 95% confidence interval for the prediction. Squares indicate the predicted response values at the selected factor levels, as generated by the statistical software.

**Figure 4 foods-14-03233-f004:**
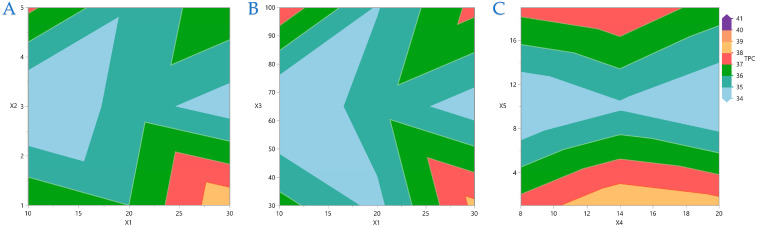
For TPC (mg GAE/g dw), plot (**A**) represents the covariation of *X*_1_ (distance between material surface and plasma source) and *X*_2_ (material thickness); plot (**B**) shows the covariation of *X*_1_ and *X*_3_ (plasma power level); plot (**C**) illustrates the covariation of *X*_4_ (nitrogen flow rate) and *X*_5_ (exposure time).

**Figure 5 foods-14-03233-f005:**
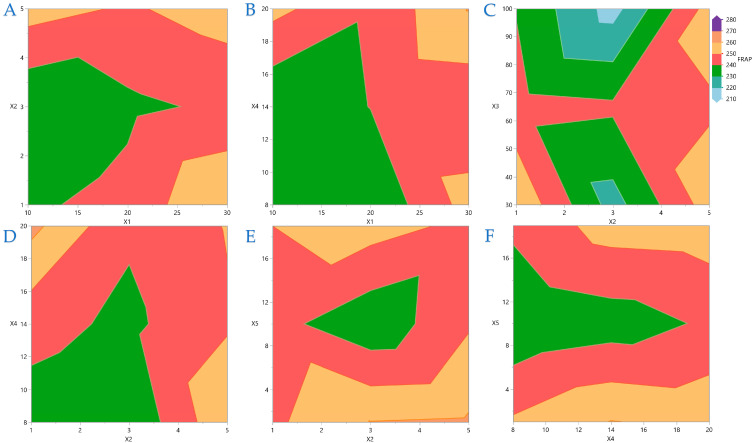
For FRAP (μmol AAE/g dw), plot (**A**) represents the covariation of *X*_1_ (distance between material surface and plasma source) and *X*_2_ (material thickness); plot (**B**) shows the covariation of *X*_1_ and *X*_4_ (nitrogen flow rate); plot (**C**) illustrates the covariation of *X*_2_ and *X*_3_ (plasma power level); plot (**D**) shows the covariation of *X*_2_ and *X*_4_; plot (**E**) presents the covariation of *X*_2_ and *X*_5_ (exposure time); plot (**F**) illustrates the covariation of *X*_4_ and *X*_5_.

**Figure 6 foods-14-03233-f006:**
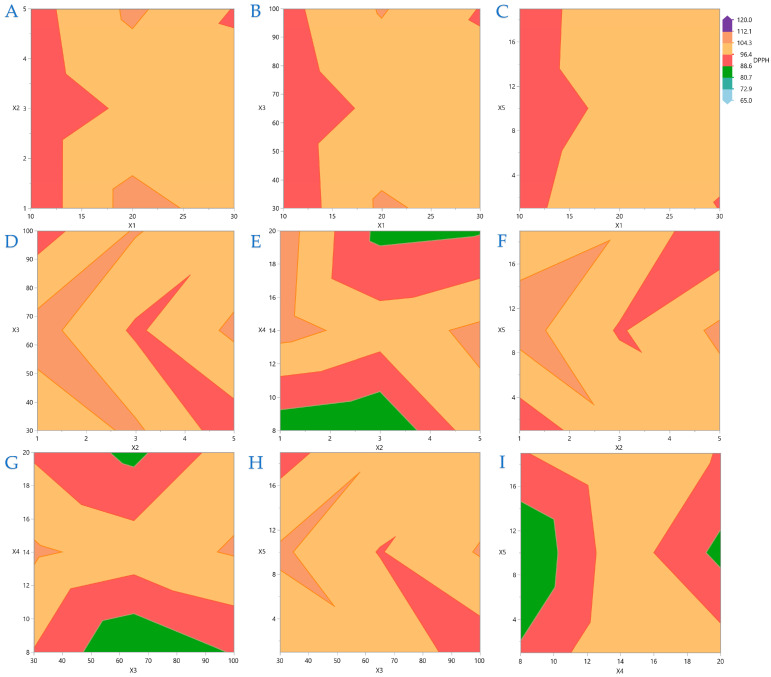
For DPPH (μmol AAE/g dw), plot (**A**) represents the covariation of *X*_1_ (distance between material surface and plasma source) and *X*_2_ (material thickness); plot (**B**) shows the covariation of *X*_1_ and *X*_3_ (plasma power level); plot (**C**) illustrates the covariation of *X*_1_ and *X*_5_ (exposure time); plot (**D**) shows the covariation of *X*_2_ and *X*_3_; plot (**E**) presents the covariation of *X*_2_ and *X*_4_ (nitrogen flow rate); plot (**F**) illustrates the covariation of *X*_2_ and *X*_5_; plot (**G**) represents the covariation of *X*_3_ and *X*_4_; plot (**H**) depicts the covariation of *X*_3_ and *X*_5_; plot (**I**) illustrates the covariation of *X*_4_ and *X*_5_.

**Figure 7 foods-14-03233-f007:**
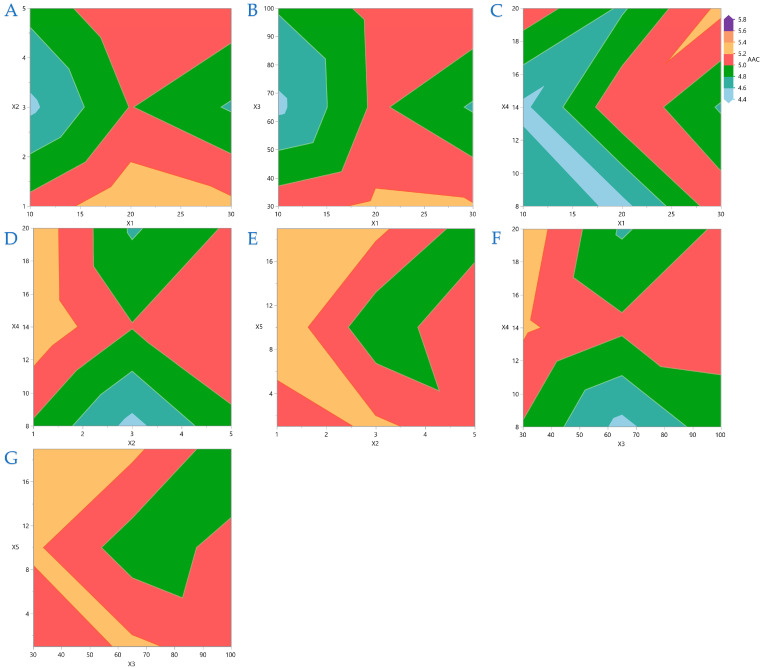
For AAC (mg/g dw), plot (**A**) represents the covariation of *X*_1_ (distance between material surface and plasma source) and *X*_2_ (material thickness); plot (**B**) shows the covariation of *X*_1_ and *X*_3_ (plasma power level); plot (**C**) illustrates the covariation of *X*_1_ and *X*_4_ (nitrogen flow rate); plot (**D**) shows the covariation of *X*_2_ and *X*_4_; plot (**E**) presents the covariation of *X*_2_ and *X*_5_ (exposure time); plot (**F**) illustrates the covariation of *X*_3_ and *X*_4_; plot (**G**) represents the covariation of *X*_3_ and *X*_5_.

**Figure 8 foods-14-03233-f008:**
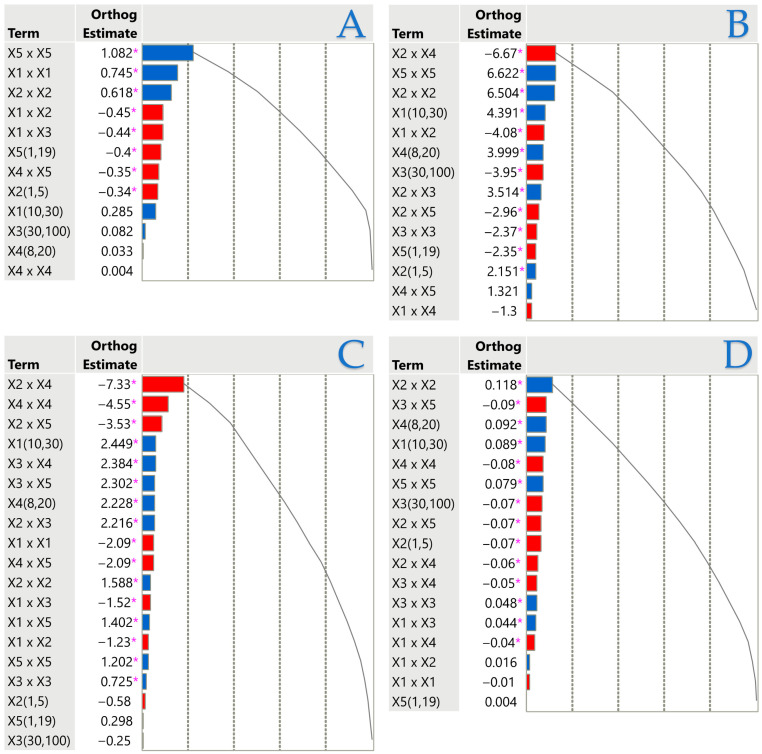
Pareto plots illustrate the significance of each parameter estimate in the ARTP pretreatment and UAE technique for TPC (**A**), FRAP assay (**B**), DPPH assay (**C**), and AAC (**D**). Significant effects (*p* < 0.05) are marked with a pink asterisk, while positive and negative contributions are represented by blue and red bars, respectively.

**Figure 9 foods-14-03233-f009:**
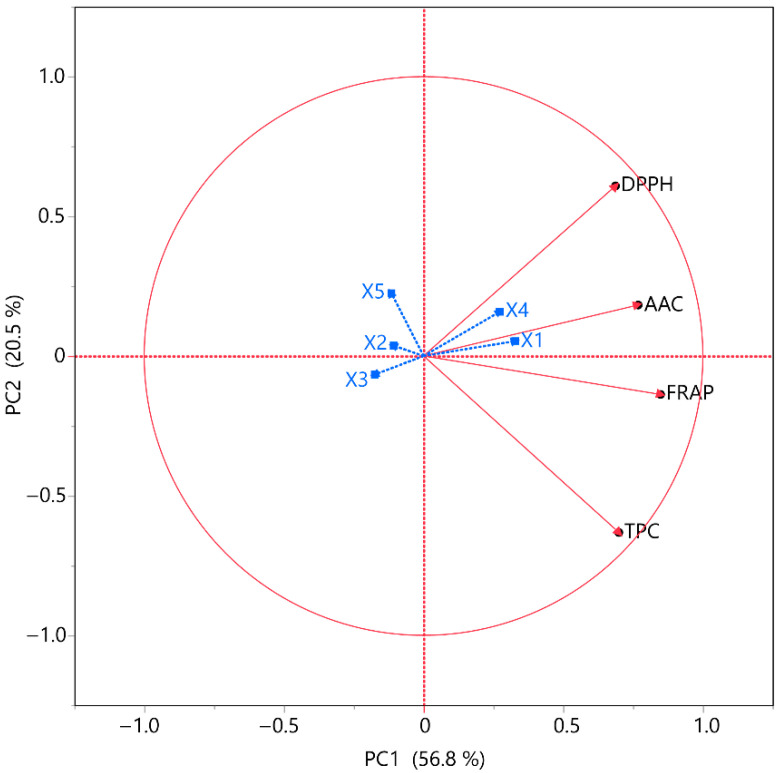
PCA for the measured variables. All *X* variables are visually encoded in blue for clarity.

**Figure 10 foods-14-03233-f010:**
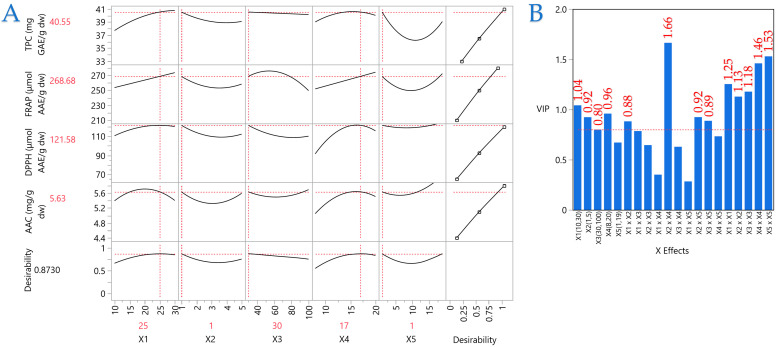
Plot (**A**) showcases the refinement of the ARTP pretreatment and UAE method for the dry MOL, employing a Partial Least Squares (PLS) prediction profiler and a desirability function with extrapolation management. Meanwhile, plot (**B**) illustrates the Variable Importance Plot (VIP), which underscores the significance of each predictor variable in the ARTP pretreatment and UAE approach. The red dashed line serves as a reference for the 0.8 significance threshold, emphasizing the relative impact of each variable within the model.

**Figure 11 foods-14-03233-f011:**
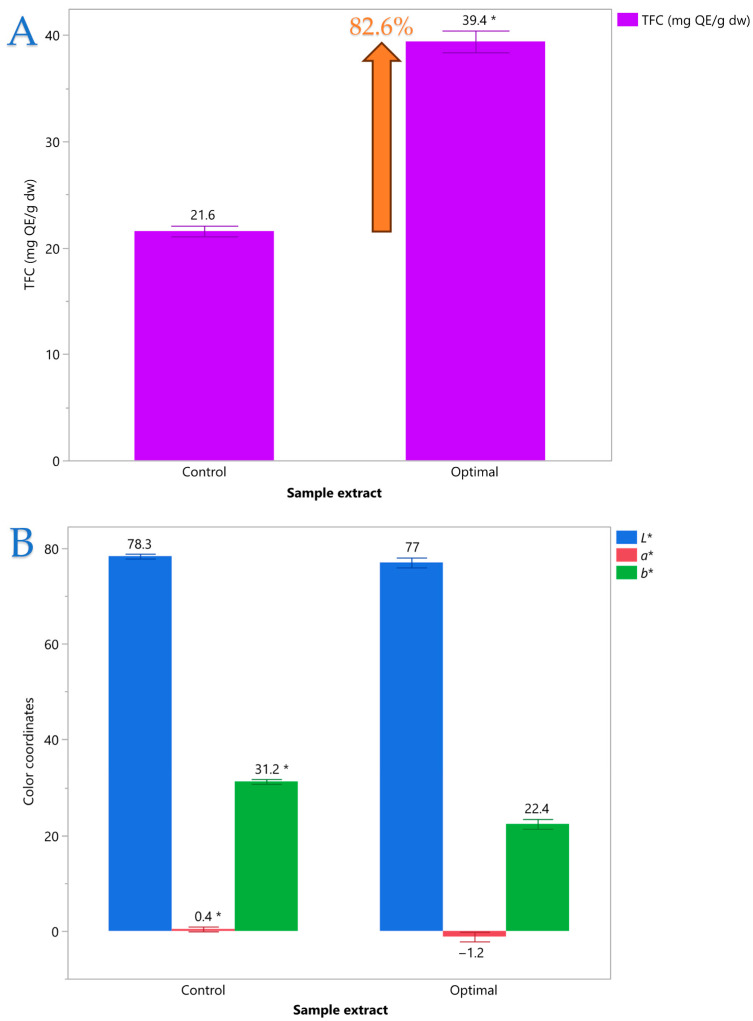
Comparative bar plot of (**A**) total flavonoid content (TFC) and (**B**) CIELAB color coordinates (*L**, *a**, *b**) in control and optimal sample extracts, with significant differences marked by an asterisk (*).

**Figure 12 foods-14-03233-f012:**
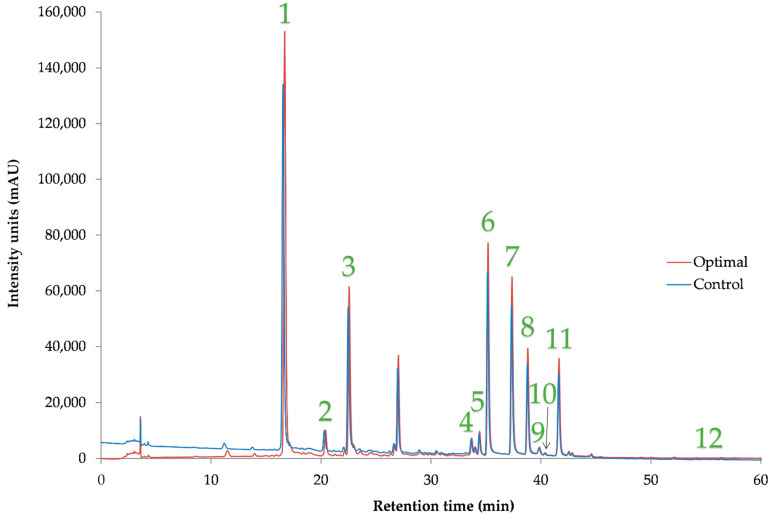
Representative HPLC chromatogram at 280 nm of the optimized dry MOL obtained through plasma treatment (optimal) and Control (without plasma treatment), illustrating the identified polyphenolic compounds. 1: neochlorogenic acid; 2: catechin; 3: chlorogenic acid; 4: ferulic acid; 5: rutin; 6: quercetin 3-*D*-galactoside; 7: quercetin 3-β-*D*-glucoside; 8: kaempferol–3-glucoside; 9: isorhamnetin–3-*O*-glucoside; 10: apigenin–7-*O*-glucoside; 11: myricetin; 12: apigenin.

**Table 1 foods-14-03233-t001:** The independent variables’ actual measurements and coded representations used during the UAE process optimization.

Independent Variables	Coded Units	Coded Variable Levels		
		−1	0	1
Liquid-to-solid ratio (*R*, mL/g)	*X* _1_	10	40	70
Ethanol concentration in water (*C*, % *v*/*v*)	*X* _2_	0	50	100
Ultrasound energy (*E*, %)	*X* _3_	60	80	100
Extraction time (*t*, min)	*X* _4_	5	10	15

**Table 2 foods-14-03233-t002:** Actual values and coded levels of the independent variables used to optimize the ARTP pretreatment.

Independent Variables	Coded Units	Coded Variable Levels		
		−1	0	1
Distance between material surface and plasma source (*D*, mm)	*X* _1_	10	20	30
Material Thickness (*T*, mm)	*X* _2_	1	3	5
Plasma power level (*P*, %)	*X* _3_	30	65	100
Nitrogen flow rate (*N*, L/min)	*X* _4_	8	14	20
Exposure time (*t*, min)	*X* _5_	1	10	19

**Table 3 foods-14-03233-t003:** The experimental and predicted results, detailing the relationships between the four independent variables examined and the corresponding responses of the TPC.

Design Point	Independent Variables				Responses	
	*X* _1_	*X* _2_	*X* _3_	*X* _4_	Actual TPC (mg GAE/g dw)	Predicted TPC (mg GAE/g dw)
1	1	−1	0	1	32.84	32.88
2	1	−1	1	0	33.39	33.39
3	−1	−1	0	−1	19.90	19.93
4	1	0	1	1	32.98	32.95
5	−0.32	0	0	0	27.30	27.39
6	−1	1	−1	−1	1.63	1.62
7	−1	0	−1	1	25.22	25.21
8	0	1	0	1	6.80	6.82
9	−1	1	1	0	0.83	0.82
10	0	0	1	−1	28.55	28.50
11	1	1	0	−1	6.75	6.79
12	1	1	−1	0	13.34	13.31
13	0	−1	−1	0	23.11	23.06
14	−1	−1	1	1	20.64	20.63
15	1	0	−1	−1	19.11	19.11

**Table 4 foods-14-03233-t004:** ANOVA summary of the quadratic polynomial model applied within response surface methodology.

Factor	TPC
Least squares regression	
Intercept	28.64 *
*X*_1_—liquid-to-solid ratio	3.788 *
*X*_2_—ethanol concentration in water	−9.04 *
*X*_3_—ultrasound energy	0.965 *
*X*_4_—extraction time	2.249 *
*X* _1_ ^2^	−0.35
*X* _1_ *X* _2_	0.955 *
*X* _2_ ^2^	−11.2 *
*X* _1_ *X* _3_	0.993 *
*X* _2_ *X* _3_	−2.46 *
*X* _1_ *X* _4_	2.711 *
*X* _2_ *X* _4_	−1.39 *
*X* _3_ *X* _4_	−3.6 *
*X* _4_ ^2^	−2.45 *
ANOVA	
*F*-value	7224
*p*-value	0.0092 *
*R* ^2^	0.9999
Adjusted *R*^2^	0.9999
RMSE	0.137
MR	19.49
CV	57.72
DF (total)	14

* At the 95% confidence level (*p* < 0.05), certain variables exerted a statistically significant effect on the observed responses. *F*-value denotes the ratio of variance explained by the model to the residual (error) variance, and the *p*-value indicates the likelihood of obtaining the observed *F*-value if the null hypothesis were true. RMSE represents the root mean square error, reflecting the standard deviation of residuals; MR is the mean response; CV expresses the coefficient of variation as a measure of relative variability; and DF signifies the degrees of freedom.

**Table 5 foods-14-03233-t005:** The experimental results, detailing the relationships between the five independent variables examined and the corresponding responses of the dependent variables. For comparative analysis, a control sample (without plasma treatment) was also included.

Design Point	Independent Variables					Actual Responses			
	*X* _1_	*X* _2_	*X* _3_	*X* _4_	*X* _5_	TPC (mg GAE/g dw)	FRAP (μmol AAE/g dw)	DPPH (μmol AAE/g dw)	AAC (mg/g dw)
Control						34.68 ± 0.59	242.16 ± 5.81	98.2 ± 3.63	4.98 ± 0.3
1	−1	−1	1	−1	1	37.19 ± 0.82	212.94 ± 6.39	83.59 ± 2.59	4.73 ± 0.22
2	0	0	0	0	−1	38.32 ± 1.61	255.53 ± 6.9	103.09 ± 3.92	5.26 ± 0.24
3	−1	−1	1	1	−1	39.03 ± 0.82	242.56 ± 10.19	102.55 ± 3.38	5.1 ± 0.23
4	1	−1	−1	−1	1	38.99 ± 0.94	256.6 ± 10.01	104.16 ± 2.5	5.32 ± 0.18
5	0	0	1	0	0	35.69 ± 0.71	218.56 ± 4.81	105.79 ± 2.75	5.15 ± 0.1
6	1	0	0	0	0	34.33 ± 0.96	246.25 ± 6.16	97.8 ± 4.11	4.98 ± 0.2
7	0	0	0	0	0	35.91 ± 1.36	236.72 ± 5.92	98.94 ± 3.66	5.14 ± 0.27
8	0	1	0	0	0	35.63 ± 1.6	250.39 ± 8.51	106.46 ± 2.66	5.57 ± 0.3
9	−1	−1	−1	1	1	35.09 ± 1.02	263.79 ± 8.44	99.84 ± 3.39	5.77 ± 0.24
10	0	−1	0	0	0	35.47 ± 1.45	243.7 ± 6.82	106.33 ± 4.57	4.96 ± 0.19
11	−1	0	0	0	0	33.87 ± 1.46	228.61 ± 7.32	90.83 ± 2.63	4.34 ± 0.17
12	−1	1	1	−1	−1	37.79 ± 1.55	267.39 ± 6.68	101.8 ± 2.75	4.88 ± 0.14
13	1	1	1	−1	1	36.96 ± 1.03	251.41 ± 8.55	104.19 ± 4.17	4.95 ± 0.2
14	1	−1	−1	1	−1	40.62 ± 1.79	279.74 ± 8.95	114 ± 4.56	5.29 ± 0.21
15	0	0	−1	0	0	33.48 ± 1.21	225.35 ± 9.24	104.29 ± 2.92	4.98 ± 0.1
16	−1	1	−1	−1	1	36.45 ± 1.13	245.42 ± 8.84	92.35 ± 2.22	4.9 ± 0.26
17	1	1	−1	−1	−1	37.38 ± 0.82	269.33 ± 8.89	108.89 ± 2.4	5.06 ± 0.17
18	0	0	0	0	0	34.39 ± 1.38	229.6 ± 5.28	101.69 ± 4.17	4.92 ± 0.21
19	0	0	0	1	0	34.82 ± 1.04	244.66 ± 5.38	88.01 ± 2.46	5.13 ± 0.18
20	0	0	0	0	1	38.43 ± 1.69	260.82 ± 6.78	103.68 ± 3.94	5.25 ± 0.28
21	−1	1	1	1	1	36.41 ± 1.06	245.66 ± 10.07	88.25 ± 3.27	4.52 ± 0.23
22	1	−1	1	−1	−1	38.76 ± 0.97	237.68 ± 9.27	67.75 ± 2.85	5.16 ± 0.14
23	0	0	0	−1	0	32.76 ± 0.66	229.35 ± 5.28	78.8 ± 1.65	4.17 ± 0.17
24	−1	−1	−1	−1	−1	36.44 ± 1.09	235.02 ± 9.4	80.72 ± 2.83	4.71 ± 0.15
25	1	1	1	1	−1	35.87 ± 1.18	250.6 ± 6.77	94.34 ± 3.02	5.22 ± 0.2
26	1	1	−1	1	1	36.59 ± 0.99	244.24 ± 10.01	76.7 ± 1.92	5.22 ± 0.2
27	1	−1	1	1	1	37.1 ± 0.82	263.89 ± 5.81	117.82 ± 2.95	5.23 ± 0.19
28	−1	1	−1	1	−1	37.93 ± 1.71	259.78 ± 10.39	90.75 ± 3.9	5.12 ± 0.26

Values are expressed as mean ± standard deviation (*n* = 3).

**Table 6 foods-14-03233-t006:** ANOVA evaluation of the quadratic model’s fit and significance in the response surface methodology.

Factor	TPC	FRAP	DPPH	AAC
Stepwise regression				
Intercept	34.83 *	236.4 *	99.52 *	4.978 *
*X*_1_—distance between material surface and plasma source	0.356	5.476 *	3.054 *	0.111 *
*X*_2_—material thickness	−0.43 *	2.683 *	−0.72	−0.09 *
*X*_3_—plasma power level	0.102	−4.92 *	−0.31	−0.09 *
*X*_4_—nitrogen flow rate	0.041	4.988 *	2.778 *	0.115 *
*X*_5_—exposure time	−0.5 *	−2.94 *	0.372	0.005
*X* _1_ *X* _2_	−0.59 *	−5.39 *	−1.63 *	0.021
*X* _1_ *X* _3_	−0.59 *	-	−2.01 *	0.059 *
*X* _2_ *X* _3_	-	4.648 *	2.931 *	−0.05 *
*X* _1_ *X* _4_	-	−1.72	-	-
*X* _2_ *X* _4_	-	−8.81 *	−9.7 *	−0.07 *
*X* _3_ *X* _4_	-	-	3.154 *	−0.07 *
*X* _1_ *X* _5_	-	-	1.855 *	-
*X* _2_ *X* _5_	-	−3.91 *	−4.67 *	−0.09 *
*X* _3_ *X* _5_	-	-	3.045 *	−0.12 *
*X* _4_ *X* _5_	−0.47 *	1.747	−2.76 *	-
*X* _1_ ^2^	−0.71	-	−5.01 *	−0.31 *
*X* _2_ ^2^	0.74	9.753 *	7.074 *	0.293 *
*X* _3_ ^2^		−15.3 *	5.719 *	0.168 *
*X* _4_ ^2^	−1.02	-	−15.9 *	−0.32 *
*X* _5_ ^2^	3.565 *	20.88 *	4.064 *	0.268 *
ANOVA				
*F*-value (model)	9.88	30.93	83	41.82
*F*-value (lack of fit)	0.585	0.587	0.572	2.624
*p*-value (model)	<0.0001 *	<0.0001 *	<0.0001 *	<0.0001 *
*p*-value (lack of fit)	0.7880 ^ns^	0.7837 ^ns^	0.7722 ^ns^	0.4477 ^ns^
*R* ^2^	0.888	0.97	0.995	0.986
Adjusted *R*^2^	0.798	0.938	0.983	0.963
RMSE	0.841	3.96	1.538	0.055
MR	36.49	246.3	96.91	5.04
CV	5.127	6.48	12.16	5.688
DF (total)	27	27	27	27

* At the 95% confidence level (*p* < 0.05), the analyzed values showed statistically significant effects on the measured responses, and ns marks non-significant results.

**Table 7 foods-14-03233-t007:** Predicted maximum responses and optimal extraction settings for each dependent variable.

Parameters	Independent Variables					Desirability	Stepwise Regression
	*X*_1_ (*D*, mm)	*X*_2_ (*T*, mm)	*X*_3_ (*P*, %)	*X*_4_ (*N*, L/min)	*X*_5_ (*t*, min)		
TPC (mg GAE/g dw)	30	1	30	17	1	0.9590	40.79 ± 1.42
FRAP (μmol AAE/g dw)	30	1	45	19	1	0.9012	283.35 ± 7.1
DPPH (μmol AAE/g dw)	24	1	100	17	19	0.9971	125.41 ± 3.31
AAC (mg/g dw)	20	1	30	17	19	0.9987	6.15 ± 0.11

Values are presented to two decimal places to maintain the precision required for design point interpretation and reproducibility of the optimization model. Rounded values are used in the main text for readability.

**Table 8 foods-14-03233-t008:** Multivariate correlation analysis revealed inter-variable relationships between TPC, FRAP, DPPH, and AAC.

Responses	TPC	FRAP	DPPH	AAC
TPC	-	0.5500	0.1983	0.3715
FRAP		-	0.4653	0.4867
DPPH			-	0.4422
AAC				-

**Table 9 foods-14-03233-t009:** The partial least squares (PLS) prediction profiler determined the maximum desirability for all variables under optimal ARTP conditions (*X*_1_: 25 mm, *X*_2_: 1 mm, *X*_3_: 30%, *X*_4_: 17 L/min, *X*_5_: 1 min) versus the Control sample (without plasma treatment).

Parameters	PLS Regression	ARTP Experimental Values	Control Values	% Increase
TPC (mg GAE/g dw)	40.55	43.29 ± 2.64 *	34.68 ± 0.59	24.8
FRAP (μmol AAE/g dw)	268.68	276.03 ± 9.94 *	242.16 ± 5.81	14.0
DPPH (μmol AAE/g dw)	121.58	113.29 ± 2.95 *	98.2 ± 3.63	15.4
AAC (mg/g dw)	5.63	5.3 ± 0.36	4.98 ± 0.3	6.3

* Values lower than 0.05 in each row are regarded as statistically significant.

**Table 10 foods-14-03233-t010:** Optimal extraction conditions for MOL polyphenolic compounds compared with the control, including percentage increase in yield.

A/A	Polyphenolic Compound	Concentration (mg/g dw)		% Increase
		Control	Optimal	
1	Neochlorogenic acid	3.53 ± 0.16	5.33 ± 0.38 *	50.8
2	Catechin	<LOQ	<LOQ	-
3	Chlorogenic acid	0.79 ± 0.03	0.94 ± 0.03 *	18.1
4	Ferulic acid	<LOQ	<LOQ	-
5	Rutin	<LOQ	<LOQ	-
6	Quercetin 3-*D*-galactoside	1.77 ± 0.12	2.37 ± 0.1 *	33.5
7	Quercetin 3-β-*D*-glucoside	0.89 ± 0.04	2.5 ± 0.06 *	180.6
8	Kaempferol–3-glucoside	0.72 ± 0.05	1.41 ± 0.08 *	96.2
9	Isorhamnetin–3-*O*-glucoside	0.77 ± 0.06	0.99 ± 0.06 *	28.8
10	Apigenin–7-*O*-glucoside	<LOQ	<LOQ	-
11	Myricetin	<LOQ	<LOQ	-
12	Apigenin	<LOQ	<LOQ	-
	Total identified	8.47 ± 0.46	13.52 ± 0.74 *	59.7

Each value is expressed as the mean ± standard deviation (SD) from three independent replicates (*n* = 3) for the respective technique. Within each row, an asterisk (*) denotes differences that are statistically significant (*p* < 0.05).

## Data Availability

The original contributions presented in this study are included in the article. Further inquiries can be directed to the corresponding author.

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
