# Peer review of "Atmospheric Room Temperature Plasma as a Green Pretreatment Strategy for Enhanced Phytochemical Extraction from Moringa oleifera Leaves"

_foods, 2025, doi:10.3390/foods14183233_

Round 1
Reviewer 1 Report
Comments and Suggestions for Authors
- In the abstract, The author mentions "Atmospheric Room Temperature Plasma (ARTP), a novel non-thermal pretreatment method has emerged as a promising green technology due to its minimal environmental impact, "cost-effectiveness,and superior extraction efficiency compared to conventional methods."? Is ARTP determined to be a low-cost extraction method? Is the price of ARTP equipment and the cost of its consumables low? All of these need to take into account the extraction cost.
- For a research study, there are 10 keywords in the manuscript, which is too many. It is recommended to keep 3 to 5 keywords.
- In Figures 1, 4, etc., the font size of the letters (A, B, etc.) is significantly larger than that of the main text, which is highly inconsistent. It is suggested that Figure 10 be divided into two figure 10A and B, to standardize the presentation of the results.
- In "2.1 Chemicals and Reagents ", it is recommended to supplement the item number of the corresponding reagent which is conducive to the reproducibility and reference value of the experiment.
- In "2.3. Instruments and Software" : This document involves the combined use of multiple devices. If a simple schematic diagram of the devices and their combined use is drawn, it will be more helpful for readers to understand.
- line 577-586: In this paragraph, the author mentioned the use of ARTP as a mutagenesis method to alter gene expression levels to achieve the goal of changing color. However, in this article, ARTP is utilized as a pretreatment method to extract active components. The same method is used for two different purposes. It is suggested to deeply explore how the use of ARTP can achieve these two different purposes. What are the similarities and differences in their principles?
- Line 620-623: “The enhanced recovery of TPC and TFC under optimal ARTP pretreatment is likely attributable to the action of reactive nitrogen species (RNS) generated by N2 plasma. These species can etch and partially oxidize cell wall polymers, structural barriers and facilitating solvent penetration ", the authors hypothesized that N2 plasma increased the recovery rates of TPC and TFC by destroying the cell wall. This can be confirmed by observing with a scanning electron microscope or a transmission electron microscope. The principle of its operation is very important. It is recommended to supplement this experiment.
Author Response
- In the abstract, The author mentions "Atmospheric Room Temperature Plasma (ARTP), a novel non-thermal pretreatment method has emerged as a promising green technology due to its minimal environmental impact, "cost-effectiveness,and superior extraction efficiency compared to conventional methods."? Is ARTP determined to be a low-cost extraction method? Is the price of ARTP equipment and the cost of its consumables low? All of these need to take into account the extraction cost.
We would like to thank the reviewer for the comprehensive revision of our manuscript.
The cost‑effectiveness of the apparatus, its low energy demand, enhanced extraction yield, and reduced solvent use are key advantages compared to other green techniques, as also discussed in citation [4]. In the revised Discussion, we have added our own operational data (0.0003 kWh electricity and 0.017 m³ nitrogen per batch) to support this claim.
- For a research study, there are 10 keywords in the manuscript, which is too many. It is recommended to keep 3 to 5 keywords.
Keywords have been reduced.
- In Figures 1, 4, etc., the font size of the letters (A, B, etc.) is significantly larger than that of the main text, which is highly inconsistent. It is suggested that Figure 10 be divided into two figure 10A and B, to standardize the presentation of the results.
We appreciate the reviewer’s observation regarding the inconsistency in the font size of the panel labels (A, B, etc.) across figures. All figure panel labels have now been adjusted to ensure a consistent font size that matches the main text for improved visual uniformity. In addition, a Figure (now included as Figure 11) has been revised and is now presented as two separate panels, Figure 11A and B, in accordance with the reviewer’s suggestion. This modification standardizes the presentation of results across the manuscript.
- In "2.1 Chemicals and Reagents ", it is recommended to supplement the item number of the corresponding reagent which is conducive to the reproducibility and reference value of the experiment.
The corresponding products numbers have been added to each chemical, as suggested by the reviewer.
- In "2.3. Instruments and Software" : This document involves the combined use of multiple devices. If a simple schematic diagram of the devices and their combined use is drawn, it will be more helpful for readers to understand.
We thank the reviewer for this valuable suggestion. To enhance clarity and reader comprehension, we have prepared a schematic diagram illustrating the combined use and sequential arrangement of all instruments described in Section 2.3. This new figure (now included as Figure 1) visually summarizes the workflow from sample preparation through ARTP pretreatment, UAE extraction, and subsequent analytical procedures. We believe this addition will make the experimental setup more accessible to readers.
- line 577-586: In this paragraph, the author mentioned the use of ARTP as a mutagenesis method to alter gene expression levels to achieve the goal of changing color. However, in this article, ARTP is utilized as a pretreatment method to extract active components. The same method is used for two different purposes. It is suggested to deeply explore how the use of ARTP can achieve these two different purposes. What are the similarities and differences in their principles?
It is clear that ARTP modifies cell wall architecture, resulting in enhanced bioactive compound yield. However, some alterations in enzymes and their bioactivity could result in chemical changes to substances that influence color or other attributes. To clarify the difference in principles, we have added a sentence in the Discussion noting that in mutagenesis applications ARTP induces genetic changes, whereas in pretreatment workflows such as the present study, the primary effect is physical and chemical modification of cell walls. A future study examining the mechanism in detail could shed more light on this hypothesis.
- Line 620-623: “The enhanced recovery of TPC and TFC under optimal ARTP pretreatment is likely attributable to the action of reactive nitrogen species (RNS) generated by N2 plasma. These species can etch and partially oxidize cell wall polymers, structural barriers and facilitating solvent penetration ", the authors hypothesized that N2 plasma increased the recovery rates of TPC and TFC by destroying the cell wall. This can be confirmed by observing with a scanning electron microscope or a transmission electron microscope. The principle of its operation is very important. It is recommended to supplement this experiment.
We agree that microscopy could provide valuable confirmation of the hypothesized cell wall disruption mechanism. While the main purpose of our work was to optimize ARTP parameters for bioactive compound extraction, in future experiments we plan to examine sample topography and crystallinity using scanning and transmission electron microscopy.
Reviewer 2 Report
Comments and Suggestions for Authors
This manuscript reports the use of atmospheric room temperature plasma as a pre-treatment safe enhancer of bioactive compound extraction. The work is overall well described and explained. However, there are some major corrections to be taken:
Line 96-97: Repetition of ideas written in lines 85-86. Also, the authors should not repeat the meaning of UAE.
Line 161: Where the leaves directly freeze-dried? They were not frozen prior to freeze-drying?
Line 162: Did the grinding was performed on heat control conditions? Or it could have occurred some sample heating?
Methods: Please describe the HPLC method used.
Please note that the results are not presented properly. The error or standard deviation should be presented with the correct significant numbers, and the result in accordance. For instance:
Line 376, 37.32 ± 2.68” should be presented as “37 ± 3”, and, in Line 377, “34.68 ± 0.59” should be presented as “34.7 ± 0.6”.
Please consider the same for further results, including Tables 7, 9 and 10.
Table 5: Were these results only calculated once? Or in triplicate? If so, please add the corresponding standard deviations.
Author Response
This manuscript reports the use of atmospheric room temperature plasma as a pre-treatment safe enhancer of bioactive compound extraction. The work is overall well described and explained. However, there are some major corrections to be taken:
We would like to acknowledge the reviewer’s positive feedback regarding our manuscript.
Line 96-97: Repetition of ideas written in lines 85-86. Also, the authors should not repeat the meaning of UAE.
The specific lines have been deleted and the following sentences have been revised.
Line 161: Where the leaves directly freeze-dried? They were not frozen prior to freeze-drying?
The leaves were kept in deep-freezing conditions (–40 °C) prior to freeze-drying process which is inserted into the manuscript.
Line 162: Did the grinding was performed on heat control conditions? Or it could have occurred some sample heating?
Grinding MOL was performed in an electric ball mill without the use of heat.
Methods: Please describe the HPLC method used.
The HPLC method is now described in the manuscript.
Please note that the results are not presented properly. The error or standard deviation should be presented with the correct significant numbers, and the result in accordance. For instance:
Line 376, 37.32 ± 2.68” should be presented as “37 ± 3”, and, in Line 377, “34.68 ± 0.59” should be presented as “34.7 ± 0.6”.
Please consider the same for further results, including Tables 7, 9 and 10.
We appreciate the reviewer’s suggestion regarding significant figures. In the main text, values are now rounded to match the significant figures implied by the associated standard deviation. In Tables 7, 9, and 10, we have retained two decimal places for both means and standard deviations to preserve the precision required for interpreting design points. A footnote has been added to each table to clarify this rationale.
Table 5: Were these results only calculated once? Or in triplicate? If so, please add the corresponding standard deviations.
All results in Table 5 were obtained from triplicate measurements (n = 3) for each experimental run. We have updated the table to present mean ±â€¯standard deviation and added a footnote clarifying the number of replicates, ensuring consistency with the reporting style of other tables.
Round 2
Reviewer 2 Report
Comments and Suggestions for Authors
I’m afraid you haven’t described the HPLC method used. You only mention the apparatus and column model. You must add the solvents used, flow and the solvent gradient used.
Author Response
I’m afraid you haven’t described the HPLC method used. You only mention the apparatus and column model. You must add the solvents used, flow and the solvent gradient used.
We thank the reviewer for this comment. We agree that the HPLC method description in the original submission could be presented more clearly. In the revised manuscript, we have reformatted and expanded the Methods section to explicitly state the solvents used, flow rate, and solvent gradient, as well as other key chromatographic parameters. The updated description now follows a standard HPLC reporting format to ensure clarity and reproducibility (see Section 2.5.1 of the revised manuscript).